# Continuous Online Action Detection from Egocentric Videos

## Abstract

Online Action Detection (OAD) tackles the challenge of recognizing actions as they unfold, relying solely on current and past frames. However, most OAD models are trained offline and assume static environments, limiting their adaptability to the dynamic, user-specific contexts typical of wearable devices. To address these limitations, we propose **Continuous Online Action Detection** (COAD), a novel task formulation in which models not only perform online action detection but also continuously learn and adapt on-the-fly from streaming videos, without storing data or requiring multiple training passes. This paradigm naturally fits egocentric vision on wearable devices, given its highly dynamic, personalized, and resource-constrained characteristics. We introduce a large-scale egocentric OAD benchmark dataset (Ego-OAD) and develop training strategies that enhance both adaptation to individual users and generalization to unseen environments. Our results on Ego-OAD demonstrate continuous learning from streaming videos improves adaptation to the user's environment by up to 20% in top-5 accuracy, and improves generalization to new scenarios by up to 7%, advancing the development of personalized egocentric AI systems.

## 1 Introduction

Wearable egocentric devices, such as smart glasses, hold promise for a wide range of real-time applications, including assistive technologies (Mucha et al., 2024) and personal AI assistants (Cai et al., 2025). A key capability for these systems is the ability to understand human actions as they unfold, directly from first-person video. Despite its importance, the majority of existing research in egocentric action understanding has focused on offline settings. In these scenarios, models are given access to the entire video sequence, including future frames, before making a prediction. While this setup is useful for post-hoc analysis or activity summarization, it is not suitable for applications that require immediate feedback. By contrast, Online Action Detection (OAD) poses a more challenging and realistic problem. In this setting, the system must recognize actions in real time, using only the current and previously observed frames, without access to future information. This constraint makes the task significantly harder, as the model must infer intent and context from partial observations, often before the action has fully unfolded.

While OAD models operate in an online manner at inference time, they are trained offline. These models are then expected to generalize properly to unpredictable input streams after deployment, but this might fail, especially in applications on wearable devices, where users, environments, and tasks vary significantly and evolve over time. Indeed, reliance on purely offline training can lead to systems that do not adapt to novel situations or personalized behaviors. To bridge this gap, we argue that OAD models must be capable of learning *on-the-fly* from continuous video streams as they are encountered in the wild, enabling real-time adaptation directly on resource-constrained devices. This capability aligns with the emerging paradigm of *on-device training*, where models continuously update using local data without relying on cloud connectivity or extensive computational resources (Zhu et al., 2024).

In this work, we introduce **Continuous Online Action Detection** (COAD), a new task formulation that enables models to not only detect actions in real time, but also train and adapt directly on continuous video streams. While COAD is broadly applicable, egocentric (first-person) video offers a particularly compelling and natural fit for this paradigm. The highly dynamic, user-centric na-

ture of egocentric video, with personalized activity patterns and constant interaction with diverse environments, demands models that can learn and adapt continuously after deployment. Moreover, the hardware constraints of wearable devices, which typically capture egocentric streams, limit the ability to store large amounts of data or to transfer those and perform costly offline retraining. These factors combine to make egocentric videos an ideal testbed for COAD.

Building on recent advances in continuous video learning (Carreira et al., 2024b; Han et al., 2025), we adapt its key principles to the OAD setting and introduce OAD-specific training strategies that enhance both *adaptation* to the user's environment and *generalization* to unseen ones. To study the COAD problem from an egocentric data perspective, we also curate a new benchmark for egocentric OAD based on the Ego4D Moment Queries (MQ) split (Grauman et al., 2022), offering a diverse and large-scale testbed for evaluating OAD models in realistic first-person settings.

In summary, our key contributions are:

- We introduce *Continuous Online Action Detection* (COAD), a new task formulation that enables models to adapt online from continuous egocentric video streams using single-pass training without the need to store data;
- We curate Ego-OAD, a new large-scale benchmark for egocentric OAD based on the Ego4D dataset (Grauman et al., 2022), providing a diverse and realistic evaluation platform for future research in this direction;
- We propose effective training strategies tailored to COAD, allowing models to specialize to individual users' environments while retaining robust generalization to new scenarios;
- We show the proposed method for COAD improves adaptation to the user's environment by up to 20% in top-5 accuracy, and boosts generalization to new scenarios by up to 7%, advancing the development of truly responsive and personalized egocentric AI systems.

## 2 RELATED WORKS

**Online Action Detection Models.**    Early research on OAD primarily focused on modeling sequential dynamics using recurrent neural networks (RNNs) (An et al., 2023; De Geest & Tuytelaars, 2018; Eun et al., 2020; Gao et al., 2017; Li et al., 2016; Xu et al., 2021a). While RNNs are efficient and well-suited for streaming video, they often struggle to capture long-range temporal dependencies, leading to degraded performance in actions that unfold over extended time windows. To address these limitations, various architectural enhancement have been proposed. Two-stream architectures (De Geest & Tuytelaars, 2018) incorporate motion cues to complement appearance features, while models such as IDN (Eun et al., 2020) and GateHub (Chen et al., 2022) introduce gating mechanisms to better control temporal information flow. Other approaches decompose the OAD task into separate modules for action recognition and action start detection, improving precision on action boundaries (Gao et al., 2017; 2019; Shou et al., 2018). More recent efforts aim to unify detection and anticipation, using either enhanced RNNs (Kim et al., 2021; Xu et al., 2021a) or Transformer-based models (Wang et al., 2021; 2023). Transformers, such as LSTR (Xu et al., 2021b) and TeSTra (Zhao & Krähenbühl, 2022), have advanced the state of the art by jointly modeling both short-term dynamics and long-term memory, enabling more accurate predictions in temporally complex scenarios (Guermal et al., 2024; Wang et al., 2023). Transformer-based models offer strong performance but incur high computational and memory costs due to their attention mechanisms, making them less suitable for real-time deployment on resource-constrained devices.

To target wearable devices deployment, in this paper, we revisit RNNs as a lightweight yet effective backbone for OAD (An et al., 2023). We demonstrate that, when equipped with an appropriate adaptation mechanism tailored for continuous video streams, RNNs achieve competitive performance in the Continuous OAD setting.

**Online Action Detection Datasets.**    Most existing benchmarks for OAD focus on exocentric video, where actions are observed from a third-person viewpoint. Datasets such as THUMOS14 (Jiang et al., 2014) and TVSeries (De Geest et al., 2016) have played a key role in advancing the field, offering challenging scenarios with diverse subjects and activity types. Nevertheless, egocentric videos are central to real-world applications involving wearable devices, such as assistive technologies and personal AI assistants. Yet, publicly available egocentric OAD datasets remain

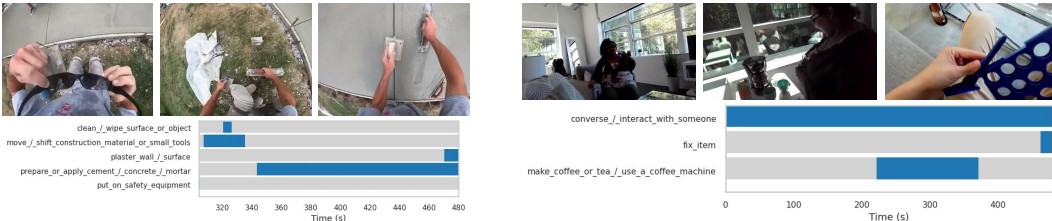

Figure 1: **Examples from Ego-OAD.** Videos sampled from the Ego-OAD, along with their corresponding multi-label, temporally grounded action annotations.

scarce, and those that do exist are often limited in either scale, diversity, or task relevance. For example, EPIC-KITCHENS (Damen et al., 2022) is a widely used egocentric dataset, but its focus on kitchen environments restricts its applicability to broader, more varied egocentric scenarios.

To bridge this gap, we introduce a new large-scale egocentric benchmark specifically designed for online action detection), curated from the Ego4D Moment Queries split (Grauman et al., 2022). Our proposed dataset captures streaming video from first-person perspectives, reflecting the temporal continuity and dynamic conditions of real-world deployment on wearable devices. It provides a more realistic testbed for evaluating models in continuous, online, and user-specific environments.

## 3    THE EGO-OAD DATASET

In this section, we detail how we curated the proposed Ego-OAD dataset from videos and annotations of the Ego4D (Grauman et al., 2022) dataset. Indeed, Ego4D (Grauman et al., 2022) contains untrimmed egocentric videos totaling 3,670 hours, collected from 8 non-US countries and 5 US states. These videos capture a wide variety of daily life scenarios (e.g., playing cards, cooking, fixing a car). The **Ego4D Moment Queries (MQ) Benchmark** evaluates temporal localization of events in long-form egocentric videos. Given a natural language query, models must retrieve the most relevant temporal segment. The MQ split covers diverse everyday scenarios with fine-grained temporal annotations and free-form query descriptions. For example, the query *"When does the person pour milk into the bowl?"* is paired with the segment from `00:01:23` to `00:01:36`, annotated with the free-form action description `pour_milk`. Although originally intended for retrieval tasks, the MQ benchmark offers rich temporal annotations that make it a strong candidate for building an OAD benchmark, enabling the study of egocentric action understanding in a realistic setting.

**Dataset Curation.**    To construct our benchmark for OAD, we curated a dataset from the untrimmed videos in the Ego4D MQ split. We treated all temporally annotated action segments as foreground instances, while unannotated intervals were considered background. Each video includes multiple annotation passes from independent annotators, who may disagree on the precise temporal boundaries or even on the action labels, reflecting the inherent ambiguity of egocentric, real-world recordings. To address this diversity, we merged all annotation passes, assigning to each frame the union of all overlapping action labels. While this strategy captures a richer range of human interpretations, it also amplifies label ambiguity: the same underlying action may be described using multiple, fine-grained categories that differ slightly (e.g., `clean_/_wipe_kitchen_appliance` vs. `clean_/_wipe_other_surface_or_object`). To mitigate this ambiguity and ensure more robust recognition, we manually grouped semantically similar free-form action descriptions into unified action classes (see Appendix A).

**Dataset Annotations.**    Ego-OAD comprises **87** fine-grained action classes and a total of **22,991** labeled action instances across **263h** of egocentric video. Videos are segmented into short clips, averaging **472s** in duration, with every frame annotated in a multi-label, temporally grounded manner. This allows for the presence of overlapping actions, i.e., **36%** of action instances partially or fully overlap with at least one other, with an average overlap duration of **9.90s**. Figure 1 shows example clips and their corresponding multi-label action annotations. We assess whether our Continuous OAD approach can enable training of an OAD model from a continuous video stream in

a deployment setting. To do this, we measure both the generalization and adaptation performance. Following the protocol proposed in Carreira et al. (2024a), we divide the data into three disjoint subsets: a **pretraining set**, an **in-stream set**, and an **out-of-stream set**, each serving a distinct role in our evaluation. Further details and statistics on the splits are described in the experimental section.

# 4 COAD: CONTINUOUS OAD

We propose an extended formulation of the standard Online Action Detection (OAD) protocol, namely *Continuous OAD (CODA)*. This extension bridges the gap between standard offline training and real-world deployment by enabling continuous model adaptation on a video stream. The following is a description of the standard OAD protocol and the key characteristics of the proposed Continuous OAD task. An overview of the method is shown in Fig. 2.

## 4.1 ONLINE ACTION DETECTION

Let $V = \{x_1, x_2, \ldots, x_T\}$ denote an untrimmed video consisting of $T$ frames, where $x_t \in \mathbb{R}^{H \times W \times 3}$ is the frame observed at time step $t$. The goal of *Online Action Detection (OAD)* is to predict a multi-label action vector $\hat{y}_t \in [0,1]^{|\mathcal{Y}|}$ for each frame $x_t$, where $\mathcal{Y}$ is the set of labels, using only visual information available up to and including time $t$. The OAD setting is subject to a strict causality constraint: the model has no access to future frames $\{x_{t+1}, \ldots, x_T\}$, but it may leverage the temporal context from the beginning of the video up to frame $t$, i.e., $\{x_1, \ldots, x_t\}$, to predict the current, potentially overlapping, action labels:

$$\hat{y}_t = f(x_{1:t}), \quad \text{where} \quad f : \left(\mathbb{R}^{H \times W \times 3}\right)^t \to [0,1]^{|\mathcal{Y}|}.$$

The overall OAD framework consists of two stages, which are described in the following.

## 4.2 STAGE 1: BACKBONE PRE-TRAINING

A video backbone $\Phi$ is pre-trained to extract local representations from short video segments. The backbone operates on temporally trimmed clips $\tilde{x} \subset V$ and produces a feature embedding:

$$z = \Phi(\tilde{x}), \quad \text{where} \quad z \in \mathbb{R}^d.$$

While different learning objectives may be used to learn useful spatiotemporal features, a common strategy is to train $\Phi$ on an offline action recognition task, where temporally trimmed input clips are labeled with action classes $y \in \mathcal{Y}$.

## 4.3 STAGE 2: OFFLINE OAD TRAINING

After pre-training, the backbone $\Phi$ is frozen and used to extract local features independently for each frame or segment:

$$z_t = \Phi(x_t), \quad t = 1, \ldots, T,$$

producing a sequence $\{z_1, z_2, \ldots, z_T\}$, with $z_t \in \mathbb{R}^d$.

The temporal detection model, typically a recurrent neural network (RNN), is then trained on sliding windows of length $\tau$ sampled from the feature sequences. During training, these windows are shuffled to obtain independent identically distributed (IID) data. For each window, the RNN predicts the action label of the last frame:

$$\hat{y}_t = f_{\text{det}}(z_{t-\tau+1}, \ldots, z_t).$$

Critically, when training on independent shuffled samples, the RNN hidden state is reset at the start of each window.

## 4.4 STAGE 3: ONLINE INFERENCE

At test time, the model performs action detection under a causal constraint. The backbone $\Phi$ extracts features $z_t$ from incoming video segments, and the detection head $f_{\text{det}}$ produces a prediction $\hat{y}_t$ based on current and past observations:

$$\hat{y}_t = f_{\text{det}}(z_1, \ldots, z_t).$$

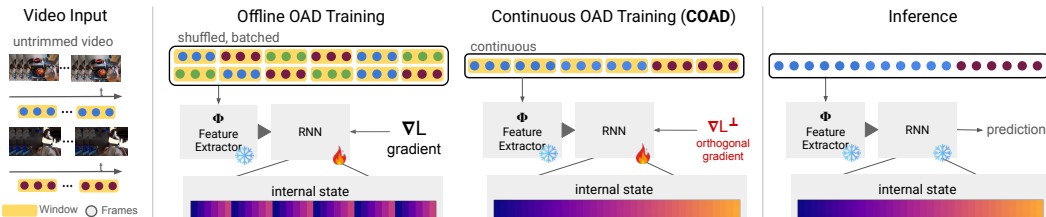

Figure 2: **Overview of COAD.** Standard offline OAD training relies on shuffled windows from the input video, resulting in IID samples. In contrast, COAD operates on a continuous video stream, handling state continuity and gradient decorrelation to enable effective training in the same streaming setting used at inference time.

Differently from training, the RNN hidden state is maintained continuously across time steps without resetting, allowing the model to leverage long-term temporal context.

## 4.5 CONTINUOUS OAD (COAD) TRAINING

We introduce an intermediate training stage between offline OAD training and online inference, enabling the model to continuous video streams.

**Single-Pass Training.** Unlike offline OAD training that relies on shuffled data, multiple epochs, and repeated access to samples, COAD operates under strict causal, single-pass constraints on temporally ordered windows. Given a continuous video stream $\{x_1, x_2, \ldots, x_T\}$, the model receives sequential windows of frames $\{x_{t-\tau+1}, \ldots, x_t\}$ at each time step $t$, where $\tau$ is the window length. Each window is processed exactly once and in temporal order. At time step $t$, the model produces a prediction $\hat{y}_t$ for the last frame in the window and updates its parameters using only information from the current and previous windows. No future frames $\{x_{t+1}, \ldots, x_T\}$ are accessible, and no replay or storage of past data is permitted. Training proceeds with batch size one over a single pass through the stream, enforcing causality and operating under tight memory and computational constraints suitable for real-time deployment.

**State Continuity.** To capture long-range dependencies from the streaming video, the temporal model maintains its hidden state continuously across frames during COAD. Using a recurrent architecture such as an RNN, the hidden state $h_t$ generally evolves as

$$h_t = \text{RNN}(z_t, h_{t-1}), \quad \hat{y}_t = f(h_t),$$

where $z_t$ is the frame-level feature from the frozen backbone. Unlike the offline training stage, which resets hidden states between shuffled windows, COAD preserves memory across all time steps. This consistency between training and inference memory states improves temporal coherence and enables effective long-term reasoning.

**Orthogonal Gradient.** Training on a continuous video stream faces the challenge of strong temporal correlations between consecutive windows, which can cause redundant or biased gradient updates. To address this, we apply an orthogonal gradient projection technique Han et al. (2025), where at each step the current gradient $g_t$ is projected to be orthogonal specifically to the gradient from the immediately preceding window $g_{t-1}$:

$$g_t^\perp = g_t - \frac{\langle g_t, g_{t-1} \rangle}{\|g_{t-1}\|^2} g_{t-1}.$$

This targeted decorrelation reduces interference between consecutive updates, allowing the model to integrate new information robustly while maintaining generalization.

**Non-uniform Loss.** In offline OAD training, RNN-based models are usually trained using sliding windows, with loss computed at each time step. Following prior work An et al. (2023), we adopt a non-uniform loss weighting strategy that computes the loss only at the final step of each window. Originally introduced to reduce the mismatch between training and inference dynamics, this

approach proves especially effective within the COAD framework, as further shown in the experimental section. Another benefit is improved label efficiency: COAD requires supervision only at each window's final step, allowing training with sparse instead of dense frame-level annotations.

## 5 EXPERIMENTS

We first evaluate models under our COAD task formulation on Ego-OAD and on the EPIC-KITCHENS dataset, which we adapt to our setting. We then present an extensive ablation analysis of COAD on Ego-OAD.

### 5.1 EXPERIMENTAL PROTOCOL

Following the protocol proposed in Carreira et al. (2024a), we divide the data into three disjoint subsets, each serving a distinct role: (1) the **pretraining set**, used for initial offline OAD training on shuffled windows with IID sampling, providing a weak initialization under limited supervision before any adaptation; (2) the **in-stream set**, used for COAD training, where the model processes continuous video in a single causal pass and updates incrementally without access to future frames or replay, simulating realistic online deployment; and (3) the **out-of-stream set**, held-out data reserved for evaluation only. On the in-stream split, we evaluate adaptation by measuring performance at each optimization step. On the out-stream split, we assess generalization after training on the in-stream data.

**Baselines.** We compare our method (**COAD**) against two reference baselines: **(1) Pretrained Only**: the model after OAD training on the pretraining set with standard IID sampling and without any further adaptation. This serves as a lower bound and reflects the model's initial performance under limited supervision. **(2) w/o COAD**: The same model trained on in-stream data without applying any of the proposed strategies, namely, orthogonal gradient regularization, non-uniform loss weighting, or state continuity.

**Datasets and Metrics.** We evaluate our approach on our proposed Ego-OAD dataset, which is designed to represent diverse scenarios of everyday activities from the egocentric perspective. We also validate our findings on the EPIC-KITCHENS-100 dataset (Damen et al., 2022), a widely-used benchmark for egocentric action understanding which focuses solely on cooking activities. For Ego-OAD, the splits consist of 186 videos for pretraining, 1,177 for the in-stream set, and 519 for the out-of-stream test, which correspond to the original Ego4D MQ validation split. We allocate the majority of training data to the in-stream split to better assess the impact of continuous learning on this data under the COAD scheme. For EPIC-KITCHENS, we evaluate COAD performance on verb, noun and action categories proposed in the original dataset (Damen et al., 2022). We split the dataset into 293 videos for pretraining, 202 for in-stream set, and 137 for the out-of-stream set. As for the evaluation metrics, we follow prior works (Zhao & Krähenbühl, 2022; Xu et al., 2021a; An et al., 2023) and report per-frame mean Average Precision (mAP), computed over all action classes, and the Top-5 Recall, which is conventionally used on EPIC.

### 5.2 IMPLEMENTATION DETAILS

Experiments on Ego-OAD use the TimeSformer backbone (Patrick et al., 2021), comparing models pretrained on either egocentric or exocentric data. For the exocentric variant, we use Kinetics-400 checkpoints (Carreira & Zisserman, 2017); for the egocentric counterpart, we use EgoVLP features (Lin et al., 2022), which apply strong egocentric pretraining on TimeSformer. Exocentric features are extracted from 8-frame clips with a stride of 2, yielding an effective rate of 1.87 FPS in both cases. We also include ablations using a TSN backbone with ResNet-50 (He et al., 2016), which processes 6-frame chunks at 24 FPS for an effective 4 FPS. For EPIC-KITCHENS, we use the official TSN features which were finetuned on the same dataset, thus reflecting egocentric pretraining only. The online detection head follows An et al. (2023), using an embedding layer, a GRU(Cho et al., 2014), and a final classifier. Training is performed in a single pass using 128-frame sliding windows with stride 16 and a learning rate of 2e-5.

Table 1: **Results on Ego-OAD.** Comparison of models with exocentric (Exo) and egocentric (Ego) pretraining for both out-of-stream and in-stream settings. The **Adaptation** column marks rows with the proposed continuous adaptation (✓). Δ represents improvements relative to the *Pretrained Only* baseline.

| Stream | Pretrain | Method | Adaptation | mAP | Top-5 Recall | Δ mAP | Δ Top-5 Recall |
|---|---|---|---|---|---|---|---|
| Out-of-stream | Ego | Pretrained Only | ✗ | 20.1 | 69.1 | — | — |
| | | w/o COAD | ✓ | 25.5 | 71.6 | 5.4 | 2.5 |
| | | COAD | ✓ | **26.0** | **76.0** | **5.9** | **6.9** |
| | Exo | Pretrained Only | ✗ | 15.8 | 55.5 | — | — |
| | | w/o COAD | ✓ | 19.0 | 57.8 | 3.2 | 2.3 |
| | | COAD | ✓ | **20.5** | **62.0** | **4.7** | **6.5** |
| In-stream | Ego | Pretrained Only | ✗ | 24.1 | 73.3 | — | — |
| | | w/o COAD | ✓ | **39.0** | 86.7 | **14.9** | 13.4 |
| | | COAD | ✓ | 36.8 | **89.3** | 12.7 | **16.0** |
| | Exo | Pretrained Only | ✗ | 15.3 | 57.5 | — | — |
| | | w/o COAD | ✓ | **31.0** | 76.2 | **15.7** | 18.7 |
| | | COAD | ✓ | **31.0** | **80.0** | **15.7** | **22.5** |

Table 2: **Results on EPIC-KITCHENS.** In-stream and out-of-stream performance **(out/in)** on EPIC-KITCHENS. Results report mAP and Top-5 Recall for verb, noun, and action. Adaptation denotes use of our proposed COAD method (✓).

| Method | Adaptation | Verb | | Noun | | Action | |
|---|---|---|---|---|---|---|---|
| | | mAP | Top-5 Recall | mAP | Top-5 Recall | mAP | Top-5 Recall |
| Pretrained Only | ✗ | 11.4 / **29.0** | 15.5 / **45.9** | 31.4 / 3.8 | 37.5 / **14.7** | 8.6 / 9.6 | **21.9 / 22.9** |
| w/o COAD | ✓ | 10.7 / 16.6 | 14.0 / 30.5 | 25.7 / 3.3 | 36.6 / 11.0 | 9.3 / 4.9 | 17.7 / 14.4 |
| COAD | ✓ | **11.8 / 29.0** | **17.0 / 45.9** | **37.1** / 3.9 | **50.2** / 13.9 | **9.9** / 7.9 | **21.9** / 20.5 |

## 5.3 RESULTS

Table 1 shows the results of our COAD method on the proposed Ego-OAD benchmark, evaluated on both the *in-stream* and *out-of-stream* splits. To assess the impact of pretraining on adaptation performance, we conduct experiments using backbones pretrained on either egocentric (ego) or exocentric (exo) data (see implementation details). The results demonstrate that egocentric pretraining significantly outperforms exocentric pretraining in both *in-stream* and *out-of-stream* settings, highlighting the critical role of egocentric representations for the Ego-OAD benchmark. COAD consistently outperforms the baseline (w/o COAD) on *out-of-stream* generalization, providing the largest gains relative to the *Pretrained Only* model before any adaptation to the continuous video stream occurs. For instance, in the egocentric setting, COAD achieves a 6.9% improvement in Top-5 Recall, compared to just 2.5% from the baseline. In the *in-stream* setting, the baseline (w/o COAD) achieves competitive results, but this often comes at the cost of reduced generalization. In contrast, COAD maintains robust performance across both domains, effectively balancing adaptation to the current stream and generalization to new, unseen data.

We also evaluate COAD on the EPIC-KITCHENS benchmark. The results in Table 1 confirm the trends observed for Ego-OAD: COAD consistently achieves the best generalization performance across all categories (Verb, Noun, and Action). On the other side, the baseline (w/o COAD) occasionally underperforms the Pretrained Only model, exhibiting signs of overfitting. In the *in-stream* setting, both COAD and the w/o COAD baseline struggle to adapt effectively. We attribute this to the fine-grained nature of the actions and annotations in EPIC-KITCHENS, which limit the model's ability to detect and exploit recurring patterns in the stream.

## 5.4 ABLATIONS

**Out-Stream vs In-Stream Trade-Off.** To better understand the trade-off between adaptation and generalization to unseen data, we analyze how performance varies in both *in-stream* and *out-stream* settings under different training hyperparameters, specifically the window stride and learning rate, as shown in Fig. 3. Higher learning rates lead the model to overfit the in-stream data, resulting in

Table 3: **Ablation on the components in COAD.** We report mAP and Top-5 Recall for both *out-stream* / *in-stream* domains. $\Delta$ denotes performance gain over the *Pretrained Only* configuration (last row).

| State Cont. | Orth. Grad. | Non-uniform | Adapt | mAP ↑ | Top-5 Recall ↑ | $\Delta$ mAP (OUT/IN) | $\Delta$ Recall (OUT/IN) |
|---|---|---|---|---|---|---|---|
| ✓ | ✓ | ✓ | ✓ | **26.0 / 36.8** | **76.0 / 89.3** | **+5.9 / +12.7** | **+6.9 / +16.0** |
| ✓ | ✓ | ✗ | ✓ | 21.8 / 42.4 | 67.7 / 88.0 | +1.7 / +18.3 | -1.4 / +14.7 |
| ✓ | ✗ | ✓ | ✓ | 25.3 / 37.4 | 71.5 / 87.9 | +5.2 / +13.3 | +2.4 / +14.6 |
| ✗ | ✓ | ✓ | ✓ | 25.9 / 36.7 | 75.8 / 89.2 | +5.8 / +12.6 | +6.7 / +15.9 |
| ✗ | ✗ | ✗ | ✓ | 25.5 / 39.0 | 71.6 / 86.7 | +5.4 / +14.9 | +2.5 / +13.4 |
| ✗ | ✗ | ✗ | ✗ | 20.1 / 24.1 | 69.1 / 73.3 | — / — | — / — |

Table 4: **Different Backbones on Ego-OAD.** We compare of frame-level and clip-level backbones under pretraining on both ego and exo datasets.

| Model | Type | Pretrain | mAP ↑ | Top-5 Recall ↑ |
|---|---|---|---|---|
| TSN | Frame | Kinetics (exo) | 17.7 | 54.5 |
| | | Ego4D MQ (ego) | 19.5 | 61.8 |
| TimeSformer | Clip | Kinetics (exo) | 26.4 | 72.8 |
| | | Ego4D (ego) | **30.0** | **82.9** |

reduced generalization capability. Conversely, increasing the window stride reduces the frequency of optimization steps performed during in-stream training, leading to degraded in-stream adaptation performance. Despite this, at higher stride values the model suffers minimal degradation in out-stream performance. Notably, at a stride of 128, the model computes the loss using a ground-truth label only once approximately every 68 seconds. This demonstrates that, under the COAD setting, the model can effectively improve performance on continuous video streams even with minimal supervision.

**COAD Components.** Table 3 presents an ablation of COAD's components, each contributing to performance. The full COAD configuration achieves the best generalization in the *out-of-stream* setting. Notably, uniform loss, effective alone, underperforms when combined with other components, while non-uniform loss boosts mAP by 4.2% and Top-5 Recall by 8.3%. Orthogonal gradient updates improve out-of-stream recall by 4.5%, highlighting the importance of gradient decorrelation in continuous video learning. State continuity provides a smaller but consistent gain, enhancing overall performance.

**Performance over Training.** COAD operates in a continuous setting: as more *in-stream* data is processed, the model improves generalization. Figure 4 shows performance evolution in the *out-of-stream* setting over time. For comparison, we include an *IID Training* baseline, where the model is trained offline with multiple passes over the combined pretraining and in-stream data, representing an upper bound under full supervision. COAD steadily narrows the gap to this upper bound, despite being limited to a single pass and online updates. Ablated variants of COAD show significantly lower performance, highlighting the importance of the full method for effective continuous learning.

**Feature Extractors.** We compare two widely used feature extractors on the Ego-OAD benchmark: TSN (frame-based) and TimeSformer (clip-based). Both are trained offline on IID samples from the combined pretraining and in-stream sets. TSN processes individual frames with late fusion, while TimeSformer captures spatio-temporal context via short clips. TimeSformer benefits from EgoVLP (Lin et al., 2022) for egocentric pretraining; since no standard egocentric checkpoint exists for TSN, we pretrain it on Ego-OAD using a standard offline action recognition setup. As shown in Table 4, egocentric pretraining improves both models, with TimeSformer significantly outperforming TSN, highlighting the value of temporal modeling in egocentric video. These results underscore the importance of adopting modern clip-based architectures for online action detection, a direction largely overlooked in prior work that has focused on frame-based models like TSN.

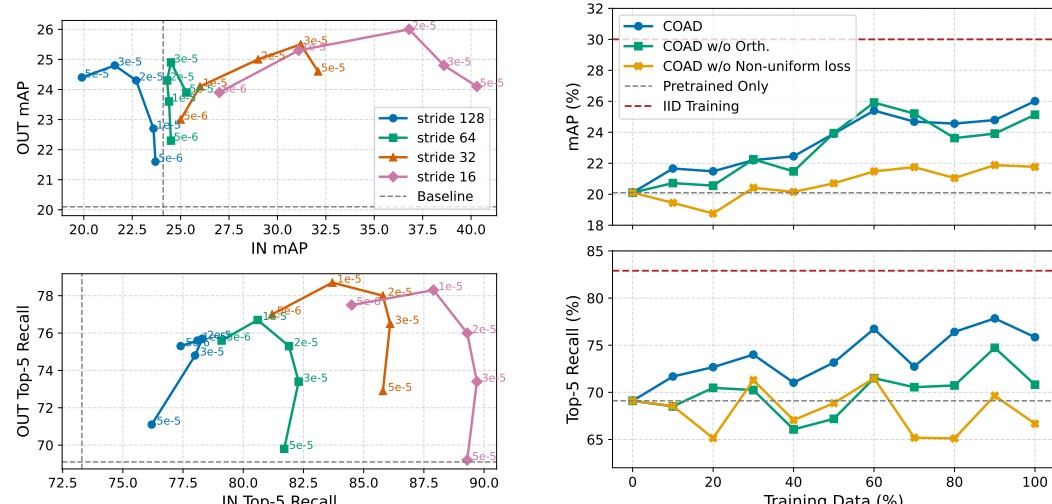

Figure 3: **In-stream *vs* out-of-stream.** Tradeoff between in-stream and out-stream performance (mAP and Top-5 Recall) as we vary the stride and learning rate.

Figure 4: **Performance on *out-of-stream* data over COAD training on in-stream data.** Performance steadily improves with more data, approaching the IID training upper bound.

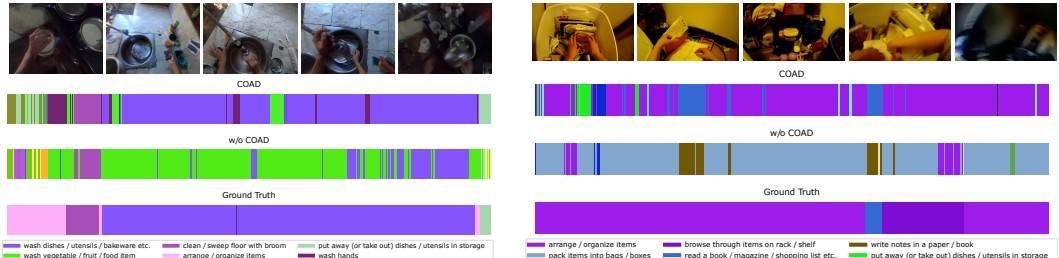

Figure 5: **Qualitative Results.** Models are trained using COAD and the w/o COAD baseline, and tested on *out-of-stream* data. For visualization purposes, only the class with the highest predicted probability is shown.

**Qualitative Results.** Fig. 5 presents per-frame predictions from COAD and the w/o COAD on two Ego-OAD videos from the out-of-stream set. As shown, COAD training on in-stream data leads to significantly better generalization.

## 6 CONCLUSIONS

We introduced Continuous Online Action Detection (COAD), a new task formulation that enables egocentric AI systems to not only recognize actions in real time, but also learn from streaming video after deployment. To support this task, we curated Ego-OAD, a large-scale benchmark derived from Ego4D featuring long-form activities in diverse environments. Our method introduces training strategies tailored for OAD from continuous video streams, aligning training with the constraints faced at inference time, and yielding significant gains in both adaptation and generalization. Experiments on Ego-OAD and EPIC-KITCHENS validate the effectiveness of COAD, establishing a foundation for responsive and adaptive first-person AI systems.

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

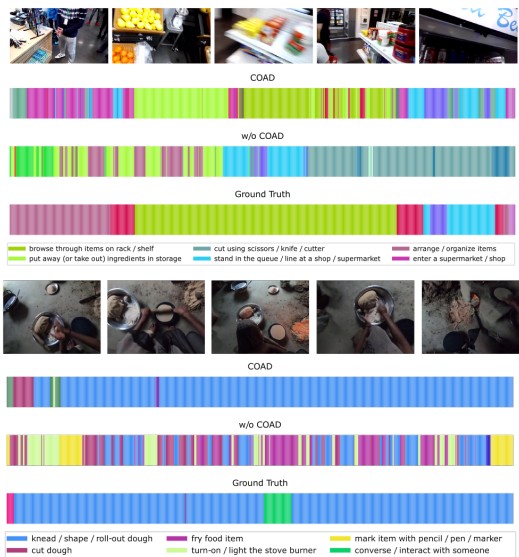

Figure 6: **Qualitative Results.** Models are trained using COAD and the w/o COAD baseline, and tested on *in-stream* data. For visualization purposes, only the class with the highest predicted probability is shown.

## A   EGO-OAD DATASET DETAILS

The Ego-OAD dataset is derived from the annotations of the Ego4D Moments Queries (MQ) split Grauman et al. (2022). To generate frame-level annotations for OAD, we treated each annotated segment as a foreground instance and assigned to each frame the union of all overlapping action labels across multiple annotation passes. To address the ambiguities introduced by multiple annotation passes and fine-grained action categories with subtle differences, we manually grouped semantically similar labels, as shown in Table 6. This aggregation resolves issues such as overlapping labels that describe supersets of each other (*e.g.*, arrange_/_organize_items_in_fridge vs. arrange_/_organize_other_items), as well as near-duplicate actions (*e.g.*, cut_tree_branch vs. trim_hedges_or_branches). Our guiding principle is that while the original fine-grained labels suit the MQ task, where the model is given a class and asked to retrieve matching segments, they are less suitable for OAD, where the model must assign labels in real time without prior hints. In this setting, subtle label distinctions can cause confusion and degrade performance, while our aggregation reduces this ambiguity making the task more robust.

Across the resulting 87 action classes, we visualize the distribution of action instances in Figure 7. The dataset exhibits significant class imbalance: while most common classes contain several thousand instances, others have only a few dozen. Figure 8 shows the average duration of action instances. Ego-OAD captures a diverse range of activities of different nature, with some spanning longer periods, such as repairing equipment or trimming grass, and others being shorter and more fine-grained, like removing food from the oven or climbing a ladder.

## B   ABLATION ON WINDOW SIZE

In our COAD framework, the model is trained on temporally ordered windows of visual features extracted from the video stream. While COAD maintains state continuity across windows, the window size determines how much temporal structure can be captured within each backpropagation step. Table 5 reports results on Ego-OAD using different window sizes during training. We find that larger windows consistently improve performance, with the best results at a size of 128, equivalent to approximately 68 seconds of video at the TimeSformer's effective rate of 1.87 FPS. This high-

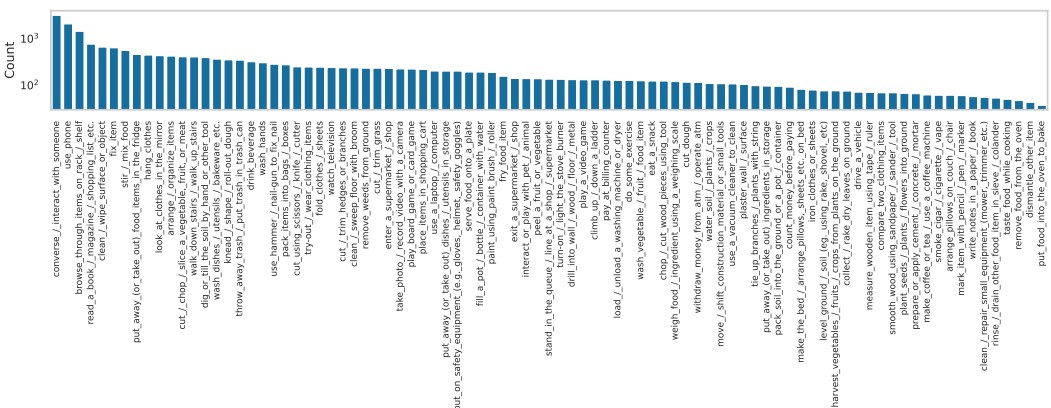

Figure 7: **Distribution of classes in Ego-OAD**, counting the individual action instances for each class.

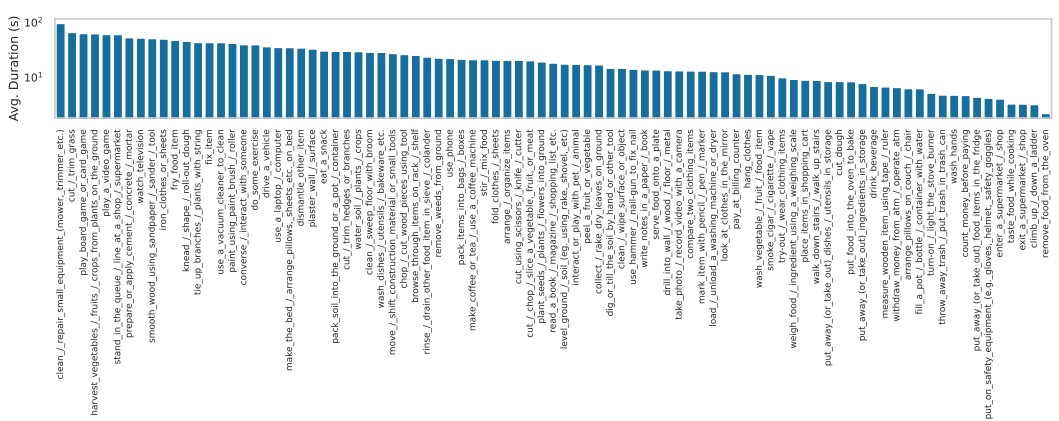

Figure 8: **Average action duration in Ego-OAD**, measured in seconds.

| Window Size | mAP ↑ | Top-5 Recall ↑ |
|---|---|---|
| 128 | **26.0 / 36.8** | **75.9 / 89.3** |
| 64 | 25.2 / 36.5 | 75.1 / 88.6 |
| 32 | 25.0 / 38.2 | 75.5 / 89.1 |
| 16 | 24.6 / 36.7 | 75.4 / 89.1 |

Table 5: **Varying sliding window size.** We report mAP and Top-5 Recall on *out-stream* / *in-stream* settings.

lights the importance of long-term temporal context for the Ego-OAD benchmark, which reflects the real-world complexity of egocentric activity streams.

## C ADDITIONAL QUALITATIVE RESULTS

Figure 6 shows additional qualitative results of COAD compare to the w/o COAD baseline on *in-stream* data. The predictions generated with COAD are markedly more stable and temporally coherent, closely aligning with the ground truth annotations. For instance, during the shopping activity, COAD sustains the label *browse through items on rack/shelf* (light green) over an extended sequence with minimal fragmentation, whereas the model without COAD exhibits erratic switching between unrelated classes such as *stand in the queue* and *cut using scissors*, despite no visual justification in the video. Notably, the transition into the activity *enter a supermarket/shop* (pink) is sharply and correctly localized by COAD, while the w/o COAD model makes a mistake in predicting this event.

Similarly, in the cooking sequence, COAD maintains a consistent prediction for *knead/shape/roll-out dough* (blue), even under variations in hand motion and illumination, unlike the baseline which fluctuates among multiple unrelated classes such as *mark item with pencil* and *converse with someone*. These results demonstrate COAD's effectiveness in suppressing noisy predictions, respecting temporal boundaries, and improving semantic fidelity, especially critical in egocentric scenarios with rapid viewpoint changes and frequent occlusions.

Table 6: **Ego-OAD label mapping.** Correspondence between original annotations in Ego4D MQ and labels defined for Ego-OAD using our semantic label aggregation.

| Ego-OAD Label | Ego4D MQ Label(s) |
|---|---|
| arrange_/_organize_items | arrange_/_organize_clothes_in_closet/dresser |
| | arrange_/_organize_items_in_fridge |
| | arrange_/_organize_other_items |
| arrange_pillows_on_couch_/_chair | arrange_pillows_on_couch_/_chair |
| browse_through_items_on_rack_/_shelf | browse_through_accessories_on_rack_/_shelf |
| | browse_through_clothing_items_on_rack_/_shelf_/_hanger |
| | browse_through_groceries_or_food_items_on_rack_/_shelf |
| | browse_through_other_items_on_rack_/_shelf |
| chop_/_cut_wood_pieces_using_tool | chop_/_cut_wood_pieces_using_tool |
| clean_/_repair_small_equipment | clean_/_repair_small_equipment_(mower,_trimmer_etc.) |
| clean_/_sweep_floor_with_broom | clean_/_sweep_floor_with_broom |
| clean_/_wipe_surface_or_object | clean_/_wipe_/_oil_metallic_item |
| | clean_/_wipe_a_table_or_kitchen_counter |
| | clean_/_wipe_kitchen_appliance |
| | clean_/_wipe_other_surface_or_object |
| climb_up_/_down_a_ladder | climb_up_/_down_a_ladder |
| collect_/_rake_dry_leaves_on_ground | collect_/_rake_dry_leaves_on_ground |
| compare_two_clothing_items | compare_two_clothing_items |
| converse_/_interact_with_someone | converse_/_interact_with_someone |
| count_money_before_paying | count_money_before_paying |
| cut_/_chop_/_slice_a_vegetable,_fruit,_or_meat | cut_/_chop_/_slice_a_vegetable,_fruit,_or_meat |
| cut_/_trim_grass | cut_/_trim_grass_with_a_lawnmower |
| | cut_/_trim_grass_with_other_tools |
| cut_/_trim_hedges_or_branches | cut_tree_branch |
| | trim_hedges_or_branches |
| cut_dough | cut_dough |
| cut_using_scissors_/_knife_/_cutter | cut_open_a_package_(e.g._with_scissors) |
| | cut_other_item_using_tool |
| | cut_thread_/_paper_/_cardboard_using_scissors_/_knife_/_cutter |
| dig_or_till_the_soil_by_hand_or_other_tool | dig_or_till_the_soil_by_hand |
| | dig_or_till_the_soil_with_a_hoe_or_other_tool |
| dismantle_other_item | dismantle_other_item |
| do_some_exercise | do_some_exercise |
| drill_into_wall_/_wood_/_floor_/_metal | drill_into_wall_/_wood_/_floor_/_metal |
| drink_beverage | drink_beverage |
| drive_a_vehicle | drive_a_vehicle |
| eat_a_snack | eat_a_snack |
| enter_a_supermarket_/_shop | enter_a_supermarket_/_shop |
| exit_a_supermarket_/_shop | exit_a_supermarket_/_shop |

| Ego-OAD Label | Ego4D MQ Label(s) |
|---|---|
| fill_a_pot_./_bottle_./_container_with_water | fill_a_pot_./_bottle_./_container_with_water |
| fix_item | fix_./_remove_./_replace_a_tire_or_wheel |
| | fix_bonnet_./_engine_of_car |
| | fix_other_item |
| | fix_pipe_./_plumbing |
| | fix_wiring |
| fold_clothes_./_sheets | fold_clothes_./_sheets |
| fry_food_item | fry_dough |
| | fry_other_food_item |
| hang_clothes | hang_clothes_in_closet_./_on_hangers |
| | hang_clothes_to_dry |
| harvest_vegetables_./_fruits_./_crops | harvest_vegetables_/_fruits_/_crops_from_plants_on_the_ground |
| interact_or_play_with_pet_./_animal | interact_or_play_with_pet_./_animal |
| iron_clothes_or_sheets | iron_clothes_or_sheets |
| knead_./_shape_./_roll-out_dough | knead_./_shape_./_roll-out_dough |
| level_ground_./_soil | level_ground_./_soil_(eg._using_rake,_shovel,_etc) |
| load_./_unload_a_washing_machine_or_dryer | load_./_unload_a_washing_machine_or_dryer |
| look_at_clothes_in_the_mirror | look_at_clothes_in_the_mirror |
| make_coffee_or_tea_./_use_a_coffee_machine | make_coffee_or_tea_./_use_a_coffee_machine |
| make_the_bed_./_arrange_pillows,_sheets_etc._on_bed | make_the_bed_./_arrange_pillows,_sheets_etc._on_bed |
| mark_item_with_pencil_./_pen_./_marker | mark_item_with_pencil_./_pen_./_marker |
| measure_wooden_item_using_tape_./_ruler | measure_wooden_item_using_tape_./_ruler |
| move_./_shift_construction_material_or_small_tools | move_./_shift_./_arrange_small_tools |
| | move_./_shift_around_construction_material |
| pack_items_into_bags_./_boxes | pack_food_items_./_groceries_into_bags_./_boxes |
| | pack_other_items_into_bags_./_boxes |
| pack_soil_into_the_ground_or_a_pot_./_container | pack_soil_into_the_ground_or_a_pot_./_container |
| paint_using_paint_brush_./_roller | paint_using_paint_brush_./_roller |
| pay_at_billing_counter | pay_at_billing_counter |
| peel_a_fruit_or_vegetable | peel_a_fruit_or_vegetable |
| place_items_in_shopping_cart | place_items_in_shopping_cart |
| plant_seeds_./_plants_./_flowers_into_ground | plant_seeds_./_plants_./_flowers_into_ground |
| plaster_wall_./_surface | plaster_wall_./_surface |
| play_a_video_game | play_a_video_game |
| play_board_game_or_card_game | play_board_game_or_card_game |
| prepare_or_apply_cement_./_concrete_./_mortar | prepare_or_apply_cement_./_concrete_./_mortar |
| put_away_(or_take_out)_dishes_./_utensils_in_storage | put_away_(or_take_out)_dishes_./_utensils_in_storage |
| put_away_(or_take_out)_food_items_in_the_fridge | put_away_(or_take_out)_food_items_in_the_fridge |
| put_away_(or_take_out)_ingredients_in_storage | put_away_(or_take_out)_ingredients_in_storage |
| put_food_into_the_oven_to_bake | put_food_into_the_oven_to_bake |
| put_on_safety_equipment | put_on_safety_equipment_(e.g._gloves,_helmet,_safety_goggles) |
| read_a_book_./_magazine_./_shopping_list_etc. | read_a_book_./_magazine_./_shopping_list_etc. |
| remove_food_from_the_oven | remove_food_from_the_oven |
| remove_weeds_from_ground | remove_weeds_from_ground |
| rinse_./_drain_other_food_item_in_sieve_./_colander | rinse_./_drain_other_food_item_in_sieve_./_colander |
| serve_food_onto_a_plate | serve_food_onto_a_plate |
| smoke_cigar_./_cigarette_./_vape | smoke_cigar_./_cigarette_./_vape |
| smooth_wood_using_sandpaper_./_sander_./_tool | smooth_wood_using_sandpaper_./_sander_./_tool |
| stand_in_the_queue_./_line_at_a_shop_./_supermarket | stand_in_the_queue_./_line_at_a_shop_./_supermarket |
| stir_./_mix_food | stir_./_mix_food_while_cooking |

| Ego-OAD Label | Ego4D MQ Label(s) |
|---|---|
| | stir_/_mix_ingredients_in_a_bowl_or_pan_ (before_cooking) |
| take_photo_/_record_video_with_a_camera | take_photo_/_record_video_with_a_camera |
| taste_food_while_cooking | taste_food_while_cooking |
| throw_away_trash_/_put_trash_in_trash_can | throw_away_trash_/_put_trash_in_trash_can |
| tie_up_branches_/_plants_with_string | tie_up_branches_/_plants_with_string |
| try-out_/_wear_clothing_items | try-out_/_wear_accessories_(e.g._tie,_belt,_scarf) |
| | try-out_/_wear_clothing_items_(e.g._shirt, _jeans,_sweater) |
| turn-on_/_light_the_stove_burner | turn-on_/_light_the_stove_burner |
| use_a_laptop_/_computer | use_a_laptop_/_computer |
| use_a_vacuum_cleaner_to_clean | use_a_vacuum_cleaner_to_clean |
| use_hammer_/_nail-gun_to_fix_nail | use_hammer_/_nail-gun_to_fix_nail |
| use_phone | use_phone |
| walk_down_stairs_/_walk_up_stairs | walk_down_stairs_/_walk_up_stairs |
| wash_dishes_/_utensils_/_bakeware_etc. | wash_dishes_/_utensils_/_bakeware_etc. |
| wash_hands | wash_hands |
| wash_vegetable_/_fruit_/_food_item | wash_vegetable_/_fruit_/_food_item |
| watch_television | watch_television |
| water_soil_/_plants_/_crops | water_soil_/_plants_/_crops |
| weigh_food_/_ingredient_using_a_weighing_scale | weigh_food_/_ingredient_using_a_weighing_scale |
| withdraw_money_from_atm_/_operate_atm | withdraw_money_from_atm_/_operate_atm |
| write_notes_in_a_paper_/_book | write_notes_in_a_paper_/_book |

## D  USE OF LARGE LANGUAGE MODELS

Large Language Models (LLMs) were used as an assistive tool to revise the writing of this manuscript (*e.g.*, grammar, phrasing). The research ideas, experiments, and conclusions are entirely our own, and the authors take full responsibility for the scientific content.

