# OpenReview forum: "Continuous Online Action Detection from Egocentric Videos"
_ICLR.cc/2026/Conference — ICLR 2026 Conference Withdrawn Submission_

### Official Review · Reviewer_31JR · 2025-10-30

**Soundness:** 3
**Presentation:** 3
**Contribution:** 3
**Rating:** 4
**Confidence:** 3

**Summary:**

The paper tackles continuous egocentric action detection and introduces Continuous Online Action Detection (COAD), emphasizing the need for models that operate on long, unsegmented video streams rather than isolated clips. To address gradient noise and distribution shifts across continuous sequences, the authors leveraged gradient orthogonalization, a Non-Uniform loss, and persistent RNN hidden states, achieving up to 6% mAP improvement on Ego4D and comparable gains on EPIC-KITCHENS. Through detailed ablations, they show that gradient orthogonalization and Non-Uniform loss are the most critical components, underscoring the difficulty of adapting exocentric models to egocentric settings.

**Strengths:**

The authors carefully designed ablation studies that show each of their modifications to a continuous (non-IID) training pipeline is effective (perhaps except for state-continuity)

The authors demonstrate a significant improvement over naive IID-pretraining across two egocentric datasets specifically for the continuous Online Action Detection task.

**Weaknesses:**

In Figures 7 and 8 from the Appendix, it appears that some categories (ex., Cutting/trimming grass, Using the phone, etc.) are both prevalent and long-lasting. If this is the case, wouldn’t reporting the per-frame mAP and Top-5-Recall, computed over all action classes, lead to an unbalanced view (by label) of the model’s performance? Wouldn’t it be fairer to weight each action class proportionally to the total number of seconds present in the out-of-stream subset? Please provide some clarifications.

Beyond merging temporal annotations from multiple reviewers and manually grouping semantically similar action classes, it is unclear what modifications the authors have made to the EGO4D MQ subset to state that their EGO-OAD dataset is a “curated” version.

**Questions:**

What are the dimensions of the embedding layer used in the RNN? Are they identical to the output embedding dimensions of the visual backbone? If so, could this alignment explain any performance differences between TSN and TimeSFormer beyond the temporal modeling effects reported in Table 4?

Regarding the ablation in Table 3, what would happen if the hidden states were not continuous and the training procedure did not involve continuous adaptation? In other words, if we maintained an IID training setup for both the visual backbone and the RNN, but retained gradient orthogonalization and the Non-Uniform loss, would we still observe comparable gains in mAP and Top-5 Recall?

It remains unclear to the reviewer that the out-of-stream setting outperforms the in-stream setup by such a large margin (over 25% mAP) for the noun classification task in EPIC-KITCHENS? If this gap is attributed to the fine-grained nature of actions and annotations in EPIC-KITCHENS, why do we not observe a similar disparity for actions and verbs in Table 2?

Additionally,1,177 videos (approximately 62%) were reserved for the in-stream subset in EGO-OAD, while only 202 videos (around 31%) were used for the in-stream subset in EPIC-KITCHENS. Is this discrepancy related to video length or another dataset-specific factor that should be clarified for the reader?

Finally, there are some other relevant methods not reported in Tables 1 and 2 for baseline comparison. For example, including some of the top-performing approaches from the Ego4D leaderboard (https://eval.ai/web/challenges/challenge-page/1626/leaderboard/3913
) could provide readers with a broader context, highlighting that continuous online action detection remains a challenging problem even for methods specifically designed for egocentric OAD.

---

### Official Review · Reviewer_KtAP · 2025-10-30

**Soundness:** 1
**Presentation:** 2
**Contribution:** 1
**Rating:** 2
**Confidence:** 3

**Summary:**

The paper introduces continuous online action detection, a new task that combined traditional online action detection with learning from continuous video streams. The paper targets this setting in egocentric video proposing a new Ego online action detection benchmark based on the Ego4D moment queries annotations. To tackle this task the paper combines orthogonal gradients with state continuity and a non-uniform loss.

**Strengths:**

- The paper addresses a timely problem: enabling models to learn continuously from streaming egocentric video. This is well motivated and relevant for wearable AI systems.
***

- The proposed method does seem to have benefit across in-stream and out-of-the-settings for both in-stream and out-of-stream settings, at least on Top-5 recall (little improvement on mAP)
***

- The Ego-4D based online action detection benchmark could be useful for future works

**Weaknesses:**

- Novelty and Positioning
- The proposed Continuous Online Action Detection (COAD) task does not appear fundamentally new. It largely combines elements of continual (or continuous) learning and online action detection, without providing a clear distinction from existing formulations.
- The paper does not clarify how much adaptation is actually required to extend existing continuous video learning methods (e.g., Carreira et al., 2024b; Han et al., 2025) to this setting, or whether these methods could serve as direct baselines.
***

- Comparison to Prior Work
- The evaluation lacks state-of-the-art baselines from both online action detection (e.g. An et al. 2023) and continuous video learning (e.g. Carreira et al. 2024b and Han et al. 2025). Without these comparisons, it is not possible to assess whether the proposed approach meaningfully advances the field.
- The main results (Table 1) instead focus more on the benefit of using egocentric data in training rather than the benefit of the proposed method itself.
- The comparison to prior work is particularly important method appears conceptually similar to earlier recurrent or memory-based OAD models, but these connections are not made explicit.
- The ablation study shows that state continuity has minimal impact, and that improvements come primarily from the non-uniform loss and orthogonal gradient components, both of borrowed from prior work. Orthoganal gradient comes from continious video work Han et al. 2025 and the non-uniform loss comes from online action detection work An et al. 2023.
***

- Dataset Contribution (Ego-OAD)
- While the dataset could be useful for future works, the contribution is relatively small since it is derived from the Ego4D momemnt queries task and annotations
***

- Evaluation Considerations
- There is no analysis of the computation–accuracy trade-off, which is critical for the proposed on-device learning setting. The cost of continuous adaptation compared to standard offline or inference-only OAD is not quantified.
- The concept of “out-of-stream” data is poorly defined. It is introduced briefly at the end of the method section and later described in the results as “held-out data reserved for evaluation only,” without a clear explanation of its role or relevance.

**Questions:**

Task Definition
- In what precise way does Continuous Online Action Detection (COAD) differ from simply combining continual learning and online action detection?
- What assumptions make COAD a distinct and necessary formulation rather than an application of existing paradigms?
***

Relation to Prior Work
- How much modification of existing continuous video learning methods (e.g., Carreira et al., 2024b; Han et al., 2025) is required to adapt them to the OAD setting?
- Why does the paper not include these prior works as baselines or points of comparison?
- How does the proposed approach differ from earlier recurrent or memory-based OAD models (e.g. An et al. 2023) and why are these works not compared to?
***

Method and Analysis Details
- Why does state continuity have such a limited impact compared to the non-uniform loss and orthogonal gradient components?
- Are these components directly adopted from prior work, or have they been substantially modified for this setting?
***

Evaluation Considerations
- What is the computational overhead of continuous adaptation during inference compared to standard OAD models?
- How significant is the accuracy–efficiency trade-off for on-device deployment?
- How is "out-of-stream" data defined and used in the experiments, and how does it differ from standard held-out test data?

---

### Official Review · Reviewer_JxkE · 2025-10-31

**Soundness:** 2
**Presentation:** 2
**Contribution:** 1
**Rating:** 2
**Confidence:** 5

**Summary:**

This paper proposes a new problem setup called Continuous Online Action Detection (COAD) for egocentric video.

Unlike standard Online Action Detection (OAD), where models are trained offline and only infer online, COAD aims to both learn and adapt continuously from streaming input, in a single-pass, without replay or multiple epochs.

The authors introduce a benchmark called Ego-OAD (a modified subset of Ego4D’s Moment Queries split) and combine three ideas to deal with the new configuration:
1. state continuity in RNNs,
2. orthogonal gradient updates (to decorrelate consecutive gradients), and
3. non-uniform loss computed only on the last frame of a window.

They claim this setting improves in-stream adaptation and out-of-stream generalization.

**Strengths:**

The proposed setup is relatively simple to implement.

**Weaknesses:**

1. **Almost no technical novelty**:
The three main components (orthogonal gradient, non-uniform loss, RNN state continuity) are all borrowed from prior work.
The proposed orthogonal gradient is directly taken from prior work on streaming learning (e.g., CVPR 2025), the non-uniform loss has already appeared in OAD literature (e.g., MiniROAD), and state continuity is essentially an inherent property of RNN-based models.
Overall, the claimed “new task and strategy” appears to be more of a reapplication or repackaging of existing ideas rather than a genuinely novel methodological contribution.

2. **Motivation is not persuasive**:
COAD assumes frame-level or interval-level action labels—extremely expensive to obtain in egocentric settings.
Labeling continuous egocentric streams for action intervals is one of the most labor-intensive tasks in video research.
If the method truly aims at continuous online learning, it should address label sparsity, delayed labels, or weak/self-supervised alternatives.
As written, it still assumes dense supervision, which fundamentally contradicts the “continuous deployment” scenario it claims to model.
In addition, the proposed formulation gives the impression of directly extending the streaming-learning setup of Carreira et al. (CVPR 2024a) into the OAD domain, without sufficiently reconciling the differing supervision assumptions and data requirements between the two tasks.

3. **Missing comparison across diverse OAD architectures**:
The paper evaluates COAD primarily on a minimal RNN-based configuration, without comparing across existing architectural paradigms.
However, most prior OAD works—such as LSTR, OADTR, MA-Transformer, GateHub, MiniROAD, and TeSTra—are fundamentally architectural contributions, largely orthogonal to the continuous-learning setup proposed here.
For the study to be experimentally complete, it is therefore important to demonstrate how COAD behaves when integrated with or compared against these architectures under a consistent backbone and feature extraction setting.

**Questions:**

Please see weaknesses.

---

### Official Review · Reviewer_9Gn1 · 2025-11-02

**Soundness:** 3
**Presentation:** 3
**Contribution:** 3
**Rating:** 4
**Confidence:** 3

**Summary:**

Traditional Online Action Detection (OAD) models are trained offline and assume static environments. This limits their adaptability to dynamic, personalized context especially in wearable devices like smart glasses. These devices capture egocentric (first-person) video streams in real time, where users, environments, and tasks vary continuously. The paper introduces Continuous Online Action Detection (COAD) to address this gap. COAD enables models to not only detect actions in real time but also learn and adapt continuously from streaming video without storing data or retraining offline. It curates a large-scale benchmark from Ego4D’s Moment Queries split for egocentric OAD with multi-label, temporally grounded annotations. It proposes training strategies of orthogonal gradient projection to reduce update redundancy, state continuity via RNNs to maintain long-term memory, and a non-uniform loss to align training with inference dynamics. In the experimental evaluation, COAD improve top-5 accuracy by 20% for in-stream adaptation and generalization performance by 7% on unseen data.

**Strengths:**

1. The paper introduces a well-motivated problem of moving beyond static, offline-trained OAD models to methods that can continuously learn and adapt in dynamic, egocentric environments.
2. The curation of the Ego-OAD benchmark provides a new, large-scale dataset for this specific task, derived from the existing Ego4D dataset.

**Weaknesses:**

1. The method/architecture novelty of the paper seems to be minimal. The proposed COAD method is a combination of pre-existing components. The orthogonal gradient technique is applied from [1] , and the non-uniform loss is applied from [2]. The third component of state continuity seems to be the default behavior of an RNN during inference. Can the authors emphasize the contributions in terms of architecture, if any?
2. The main results in Table 1 show that COAD method improves out-of-stream generalization (26.0 mAP) but worsens in-stream adaptation (36.8 mAP) compared to the "w/o COAD" baseline (25.5 mAP and 39.0 mAP, respectively, with Ego pretraining). Can the authors discuss why the performance in generalization task improves but seems to be less for in-stream data?

[1]. Han, Tengda, et al. "Learning from Streaming Video with Orthogonal Gradients." Proceedings of the Computer Vision and Pattern Recognition Conference. 2025.
[2]. An, Joungbin, et al. "Miniroad: Minimal rnn framework for online action detection." Proceedings of the IEEE/CVF International Conference on Computer Vision. 2023.

**Questions:**

Suggestions:
The text in Section 5.3 refers to Table 1 for Epic-kitchens-100 results but these results are located in Table 2.

---

### Note · Authors · 2025-11-17

I have read and agree with the venue's withdrawal policy on behalf of myself and my co-authors.